# Trace Element Speciation and Nutrient Distribution in *Boerhavia elegans*: Evaluation and Toxic Metal Concentration Across Plant Tissues

**DOI:** 10.3390/toxics13010014

**Published:** 2024-12-26

**Authors:** Tahreer M. Al-Raddadi, Lateefa A. Al-Khateeb, Mohammad W. Sadaka, Saleh O. Bahaffi

**Affiliations:** 1Chemistry Department, Faculty of Science, King Abdulaziz University, Jeddah 21589, Saudi Arabia; tmradade@uqu.edu.sa (T.M.A.-R.); laalkhatib@kau.edu.sa (L.A.A.-K.); 2Chemistry Department, Al-Qunfudah University College, Umm Al-Qura University, Makkah 24382, Saudi Arabia; 3King Fahd Medical Research Center, King Abdulaziz University, Jeddah 21589, Saudi Arabia; 4College of Health Technology, Cihan University-Erbil, Kurdistan Region, Erbil 44001, Iraq; waleed.sadaka@gmail.com

**Keywords:** *Boerhavia elegans*, chemical speciation, essential and toxic elements, medicinal plants, plant tissues, PCA

## Abstract

This study investigated the elemental composition of *Boerhavia elegans*, addressing the gap in comprehensive trace element profiling of this medicinal plant. The research aimed to determine the distribution of macronutrients, micronutrients, and beneficial and potentially toxic elements across different plant parts (seeds, leaves, stems, and roots). Using ICP-OES analysis, two digestion methods were employed to capture both complex and labile elements. The study revealed distinct elemental distribution patterns, with iron and nickel concentrating in stems, manganese and zinc in leaves, and copper in roots. Magnesium emerged as the most abundant macronutrient, particularly in leaves. Importantly, all detected toxic elements (arsenic, chromium, lead, and cadmium) were below WHO safety limits. These findings provide crucial insights into the nutritional and safety profile of *B. elegans*, potentially informing its use in traditional medicine and highlighting its potential as a source of essential elements.

## 1. Introduction

In the last century, there has been an increasing global concern over soil trace element contamination, which has coincided with an escalation of economic and societal activity. Anthropogenic sources are widely recognized as the primary driver of elevated soil trace element levels. Trace elements, including Cu, Ni, and Zn, are crucial for agricultural productivity and environmental health. They play a vital role in maintaining the quality of crops, food, and consumer health. These elements are required by plants in specific, small quantities. Their presence in the soil directly impacts the nutrient balance in animal feed (fodder). However, exceeding these recommended trace element levels can be detrimental, causing toxicity in plants and soil organisms. Therefore, maintaining optimal trace element availability in soil is essential for sustainable agriculture [1]. Edible and medicinal plants are substances with biological activity that have been used for millennia to cure many human ailments due to their low adverse effects [2,3]. However, some medicinal plants and their mixtures may pose health risks owing to toxic elements. The contamination may result from environmental pollution [4]. Creating, regulating, and using herbal or traditional medicines in many parts of the world can be challenging. Regulatory status, quality control, safety and efficacy assessment, safety monitoring, and a need for more awareness about conventional, complementary, and alternative medicines among public drug regulatory bodies are common issues in many nations [5]. Numerous mineral elements present in plants are necessary for human nutrition [6]. The literature extensively documents that they possess a range of metabolic functions, from cell defense to primary and secondary metabolism [7]. In addition to being necessary for forming active compounds in medicinal plants, trace metals also cause toxicity in those plants [8].

Trace elements are essential in plant metabolism, forming chemical constituents in plants and acting as cofactors for enzymes [9]. The contents of major and trace elements in plants are governed by the geochemical features of the soil where the plant grows and by the ability of plants to accumulate elements selectively. Moreover, plants can accumulate elements from the surrounding aerial or aquatic environment, enabling some plants to be used as biomonitors [10,11,12]. The concentration of toxic heavy metals in plants can be influenced by the geochemical parameters of the soil, air, and water pollution, as well as the ability of plants to accumulate certain elements selectively [13]. Metals may also be related to the geographical origin, harvesting, or gathering of these plant materials. Plants serve as an essential medium for trace elements to transit from the soil to humans [14]. Consequently, quality control measures for trace element content in these medicinal plants are necessary.

“Heavy metals” are metals with a density greater than 5 g/cm^3^. This group contains more than 60 metals. The primary sources of heavy metals are mineral fertilizers, some base rocks, sewage sludge, pesticides, wastewater, urban wastes, motor vehicle exhaust gases, and mining [13]. Because of their nature, these elements are typically found in the earth’s crust as oxide, carbonate, sulfides, and silicates or in stable compounds. All living creatures require some heavy metals in their surroundings to continue their daily activities. Elements that plants need to survive are called “plant nutrients”. Analyzing plant tissues reveals almost all the elements found in nature. Although plants are selective about taking up nutrient ions, as the concentration of nutrient forms of elements increases in the growth medium, some non-essential heavy metals can passively enter the plant body and enter the food chain [15]. Some metals, such as iron (Fe), copper (Cu), manganese (Mn), cadmium (Cd), molybdenum (Mo), silicon (Si), and boron (B), are essential heavy metals required for plant metabolism. Cobalt (Co), copper (Cu), iron (Fe), manganese (Mn), Molybdenum (Mo), zinc (Zn), selenium (Se), and iodine (I) are also essential heavy metals needed for animals. However, some metals can be toxic at high concentrations [13]. Various heavy metals (HMs), such as arsenic (As), cadmium (Cd), chromium (Cr), lead (Pb), and mercury (Hg), cause severe toxicity in plants when they enter agricultural soil ecosystems through human activities or natural processes [16].

Medicinal plants are an ancient practice that dates to historical times. They are widely utilized across the globe and are prevalent in many countries. Over 70% of the world’s population uses medicinal plants to treat various ailments. However, medicinal plants can also pose a risk of exposure to toxic elements, depending on their origin and characteristics [17]. Plants are crucial in transferring trace elements from the soil to humans [14]. Analyzing the metal ion compositions of medicinal plants is essential for understanding their medicinal, nutritional, and toxic properties [18]. With increasing awareness about elemental composition, elemental analysis is becoming more significant. Consequently, there is a growing interest in developing rapid, simple, and sensitive methods that can be used for routine analysis. Sample preparation methods are typically critical in chemical analysis. Acid digestion, the most commonly employed sample treatment for determining elemental concentration in biological materials like plants, necessitates high energy input and concentrated mineral acids [19]. Quantifying trace elements in medicinal plants is important for several reasons. It aids in the treatment of various diseases by understanding the pharmacological effects of these plants. It also helps determine the appropriate dosage of herbal medicines derived from medicinal plants. Furthermore, it enables the evaluation of the effect of medicinal plants on human health [8].

The plant genus *Boerhavia* has increasingly interested physiochemists as past studies have demonstrated its wide range of pharmacological and biological effects, signifying potential therapeutic applications. Research has found that a methanolic extract from *B. diffusa* leaves exhibits antibacterial properties and noteworthy inhibition against Staphylococcus aureus, a known pathogenic bacterium [20]. According to traditional medicine systems, *B. elegans* has been used to treat various other health conditions, including painful menstruation, urinary tract infections or other issues, intestinal infections, inflammation, jaundice, and overall weakness or fatigue [21]. *Boerhavia elegans* plant, as part of the plant kingdom, requires various mineral elements for growth, development, and reproduction. These mineral elements are mobilized from the soil matrix and absorbed by the roots as mineral ions.

To the best of our knowledge, no study has been reported regarding trace metal concentration and speciation of selected *Boerhavia elegans* tissues. Therefore, the current research is focused on I- determining the total trace metal concentration in various tissues and II- speciation of labile and complex fractions of selected elements. The concentrations of trace elements in the root, stem, leaf, and seed of the *Boerhavia elegans* plant were determined using the inductively coupled plasma optical emission spectroscopy (ICP-OES) technique.

Several techniques have been used for the quantity of trace elements in food samples, such as the ICP-OES, fame atomic absorption spectrometry (FAAS), and graphite furnace atomic absorption spectrometry (GFAAS) [22,23,24,25]. Among these techniques, ICP-OES was reported as the most sensitive method, time-saving, and specificity [26,27,28].

## 2. Materials and Methods

### 2.1. Sampling

Seeds, stems, leaves, and roots of *Boerhavia elegans* were analyzed in this study. The plant seeds of *B. elegans* were purchased from a local market in Jeddah, Saudi Arabia, as they are used in preparing porridge, a famous food in the Arabian Peninsula. The roots, stems, and leaves were collected from the same field plots in Hadramout, Alborgat, South Yemen. Samples were cleaned with tap water and deionized water, then air-dried in an oven (JS Research Inc., Gongju, Republic of Korea) at 50 °C. The dried samples were ground with an electrical grinder and sieved through a 100-mesh standard sieve. All the samples were initially stored in closed plastic bags until analysis.

### 2.2. Reagents and Chemicals

A multi-elemental standard solution of 1000 mg/L containing all analyzed elements (Cd, Co, Cr, Cu, Fe, Mg, Mn, Ni, Pb, Al, As, Se, Zn, K, Na, B, Ca, and P) supplied by Merck (Darmstadt, Germany) was used for calibration. Analytica grade HNO_3_ 65% and H_2_O_2_ 30% from Merck (Darmstadt, Germany) analytical grade were used for sample digestion. Ultrapure water delivered by a Milli Q system (Millipore, Merck Millipore, France) was used to dilute and prepare a more diluted and amore-diluted working solution.

### 2.3. Instrumental Analysis

A PerkinElmer Optima 7000 DV ICP-Optical Emission Spectrometer (Shelton, CT, USA), equipped with Win Lab 32 for ICP Version 4.0 software, was utilized to measure the levels of Cd, Co, Cr, Cu, Fe, Mg, Mn, Ni, Pb, Al, As, Se, Zn, K, Na, B, Ca, and P. The ICP-OES was set with a wet plasma aerosol type and an axial view, using a nebulizer startup instant condition that included a flow rate (Ar) of 15 L/min, an auxiliary flow (Ar) of 0.2 L/min, a nebulizer flow (Ar) of 0.8 L/min, a sample uptake rate of 1.5 mL/min, and a sample flush time of 5 s. Calibration was performed using a standard mixture with concentrations of 1, 5, 10, and 15 µg/mL for each element, ensuring high accuracy and precision (%RSD).

### 2.4. Recommended Procedures

#### 2.4.1. Wet Digestion

*Boerhavia elegans* seeds, leaves, stems, and roots were washed with tap water and deionized water and dried at 100 °C. for 24 h until they achieved a constant weight to remove moisture. The plant parts were ground to fine particles using an electrical grinder. An accurate mass (10.0 ± 0.001 g) (except for seeds and leaves, 5.0 g was taken) of each part of the plant sample was digested in a 100 mL beaker containing 50 mL HNO_3_ (69%). The beaker was covered with a watch glass and left overnight at room temperature, followed by the addition of 12.5 mL of H_2_O_2_ 30% (Sigma-Aldrich, Gillingham, UK), and heated on a hot plate to 110–120 °C until the light brown-colored fume disappeared. The volume became nearly dry, and 5 mL of deionized water was added. The digest solutions were filtered through Whatman No. 42 filter paper and diluted to 50 mL using HNO_3_ (0.1 M). The concentration of elements was finally measured.

#### 2.4.2. Analysis of Labile (Water-Soluble) Trace Elements in *B. elegans* Tissues

Parts of different plant tissues (seeds, leaves, stems, and roots) were dried at 100 °C until they achieved a constant weight to remove moisture and then homogenized using an electric grinder. An accurate mass (10.0 ± 0.001 g) of dried *B. elegans* tissues was transferred to deionized water (20 mL 18.2 MΩ·cm^−1^) and left overnight in a shaker at room temperature. The beaker was covered with a watch glass and heated on a hot plate to 90–120 °C for 30 min. Sample solutions were vacuum-filtered through the Whatman No. 2 filter paper. The filtrate and washing solutions were transferred to a 50 mL volumetric flask, completed to the mark using deionized water and stored in Low-Density Polyethylene (LDPE) bottles. An appropriate volume of the *B. elegans* infusion was filtered through a 0.45 µm membrane filter, and the labile water-soluble elements were analyzed versus blank solution. The ICP-OES signal intensity (I_1_) for *B. elegans* is a measure of labile concentration (C_L_) of the element in the extract in boiling water, which was determined with the help of a consistent calibration plot employing the equation:(1)$$\mathrm{Average}\;C_{L}\;µg\;g^{-1}=(C-B)\times \frac{V}{m}$$
where C is the element concentration (µg mL^−1^) of *B. elegans* tissue, B is the blank result, V is the sample volume (mL), and m is the mass (g) of the *B. elegans* tissue sample.

### 2.5. Statistical Method

For statistical analysis, XLSTAT Software 26.4.0 was used to determine PCA techniques.

## 3. Results and Discussion

### 3.1. Trace Metals in Plant

Trace metal concentrations in the seeds, leaves, stems, and roots of *B. elegans* in Table 1. The results indicate variable metal levels in different types and parts.

In *Boerhavia elegans*, ten essential elements, including macronutrients (potassium, calcium, magnesium, phosphorus) and micronutrients (iron, manganese, zinc, copper, boron, nickel), along with aluminum (not essential for plants), beneficial elements (sodium, selenium, cobalt), and potentially toxic elements (cadmium, lead, chromium, arsenic) were analyzed, with the latter posing risks to plant physiology.

Determining the distribution and speciation of metals in plants is essential for understanding the mechanisms underlying their toxicity to plants. In this study, the total concentrations of micro- and macro-elements in *B. elegans* were found to exceed the critical deficiency levels that are known to limit plant growth and survival. Table 2 presents the statistical analysis of metal concentrations in the seeds, leaves, stems, and roots of *B. elegans*.

The data reveal distinct differences among the four groups. Seeds exhibited the lowest average metal concentration (115.75) but had the highest variability, with a variance of 185,790, indicating significant differences within this group. Leaves and stems recorded the highest average concentrations, nearly identical at 293.88 and 294.16, respectively. However, leaves displayed the highest variability (1,146,789), suggesting substantial internal differences, while stems showed lower variability (653,386), reflecting greater consistency. Roots had a moderate average concentration of 178.25 and the lowest variance (239,611), indicating the most uniform data among the groups.

Overall, while leaves and stems have comparable average concentrations, leaves exhibit significantly higher variability. Roots are the most consistent group, and seeds, despite their low average concentration, demonstrate considerable variability. These findings highlight the tissue-specific distribution and variability of metal concentrations in *B. elegans*.

### 3.2. Toxic Elements

The presence of hazardous elements such as arsenic (As), cadmium (Cd), and lead (Pb) in contaminated soils poses a risk to crop plants and, ultimately, human health. The World Health Organization (WHO) has set limits for the concentration of these heavy metals in wild plants, specifying maximum levels of 5 mg/kg for arsenic, 10 mg/kg for lead, 0.3 mg/kg for cadmium, and 2 mg/kg for chromium [29].

This study investigated the distribution of cadmium, lead, arsenic, and chromium in different parts of the *B. elegans* plant. The results indicate that seeds contain the lowest levels of cadmium, while leaves exhibit the highest concentration, as illustrated in Figure 1 and Table 3. Lead concentrations are highest in leaves and lowest in seeds. Arsenic concentrations, on the other hand, are highest in roots and lowest in leaves. The concentration of arsenic in seeds ranges from 5.34 µg/g to 3.71 µg/mL. Chromium concentration shows a more balanced distribution across all plant tissues, with the highest concentration observed in seeds. Toxic metals enter root cells and are moved to other plant tissues via membrane transporters that typically handle essential or beneficial nutrients. This occurs because of the physicochemical similarities between toxic metals and nutrients, especially those within the same periodic table group, and the versatility of transporter proteins [30].

Cadmium toxicity induces the overproduction of reactive oxygen species (ROS), resulting in damage to plant membranes and the breakdown of cellular biomolecules and organelles. This toxicity also impairs the uptake of crucial nutrients like Fe and Zn, leading to leaf chlorosis. Cadmium also obstructs the transportation and absorption of vital minerals such as K, Mg, Ca, P, and Mn. Cd is primarily absorbed and transported within plants via transporters designed for divalent cations such as Fe (II), Mn, and Zn. In Arabidopsis, both the Fe (II) transporter IRT1 and the Mn transporter NRAMP1 can transport cadmium, significantly contributing to its uptake in the plant [31].

Lead (Pb) toxicity poses a significant threat to plants, animals, and humans due to its harmful effects. The rise in industrial activities involving Pb and the use of Pb-containing products like agrochemicals, oil, paint, and mining operations have led to environmental contamination, allowing Pb to infiltrate the food chain. As one of the most dangerous heavy metals, lead’s presence in the food chain represents a serious health risk for both plants and humans [32]. Inorganic lead (Pb) is commonly found in dust, soil, old paint, and various consumer products, whereas organic lead, such as tetraethyl Pb, is mainly present in leaded gasoline. Both forms are toxic, but organic Pb complexes are particularly harmful to biological systems, posing a much greater threat than their inorganic counterparts [32,33]. Plants absorb free Pb ions either through capillary action or from the atmospheric air via cellular respiration. Once lead is absorbed into the soil from the external atmosphere, it enters the plant system. Through their well-developed root systems, plants absorb nutrients from the soil, including divalent-free Pb cations in contaminated soil. These absorbed lead ions are subsequently transported through the xylem vessels [34].

Arsenic (As) contamination in the environment poses a substantial global concern impacting environmental, agricultural, and health sectors because of its highly toxic and carcinogenic nature. Even at minimal concentrations, exposure of plants to As can trigger a wide range of morphological, physiological, and biochemical alterations [35]. Arsenic is considered non-essential for plants and other organisms [36]. Plant uptake of arsenic depends on its total concentration and, importantly, on its speciation in soil. In plants, arsenic primarily enters in the form of inorganic species, As (III) or As (V), via transporter proteins regulated by the concentration gradient between the growth medium and plant cells [37]. Xylem tissues facilitate the translocation of arsenic from roots to shoots and its distribution across various plant tissues. Arsenate competes with phosphate (Pi) for uptake through Pi transporters [38]. Once inside plant cells, As (V) is enzymatically reduced to As (III) by arsenate reductase (ACR2). Detoxification of As (III) involves its complexation with thiol-rich peptides, leading to its sequestration in vacuoles of root cells, which limits As (III) efflux and enables its long-distance transport to other plant tissues [39].

While chromium trioxide (Cr (III)) is essential for animal and human health in small amounts, plants don’t require it. Chromium (Cr) contamination has become a major environmental concern due to its high levels in various agricultural and industrial activities [16]. Significant amounts of Cr are mined or produced annually, which contributes to its presence in soil and water. Natural sources of Cr include rocks, volcanic dust, gases, soil, animals, and plants, and Cr is often associated with primary rock-derived phases and well-crystallized iron oxides. Additionally, natural leaching from rocks and topsoil leads to substantial levels of Cr in water bodies [40].

Chromium (Cr) toxicity in plants adversely affects physiological, biochemical, and molecular traits, resulting in stunted growth and reduced yield. High Cr accumulation impacts seed germination, root and shoot growth rates, and overall biomass and yield. It inhibits photosynthesis, disrupts the cell cycle, and affects water and mineral balance, enzyme activity, nitrogen assimilation, the antioxidant system, and other metabolic processes. Cr accumulation also triggers the generation of reactive oxygen species (ROS), causing oxidative damage and cellular component damage, leading to cell death and altered morphology.

Because chromium (Cr) easily reacts with oxygen in the air, it’s a very active metal. Cr can exist in multiple chemical states, ranging from zero to positive six. In nature, the most common forms are chromium trioxide (Cr (III)) and chromium hexoxide (Cr (VI)). However, Cr (VI) is more dangerous because it dissolves and moves around more easily in water [41]. Plants can absorb both types of chromium [42]. Cr (VI) actively gets into plant cells through specific channels, while Cr (III) enters more passively through exchange sites on cell walls [43]. Additionally, certain acids released by plant roots help dissolve Cr, making it easier for plants to take it up [41].

### 3.3. Macronutrient Elements

Levels of the macro (P, Mg, K, and Ca) elements in plant samples are given in Table 4, Figure 2.

The concentration of magnesium (Mg) in plants was observed to range between 1839.0 and 5198.0 µg/mL. Among different plant parts, leaves exhibited the highest total Mg concentration and readily available Mg (labile Mg), followed by seeds, stems, and roots, emphasizing the critical role of Mg in leaves, where it can exceed 70% of the total dry weight. Mg forms soluble complexes with organic anions like malate, citrate, and oxalate, facilitating its mobility and distribution within plants. Unlike calcium (Ca), Mg does not substitute for Ca in the diffusible fraction when Ca levels are low. Interestingly, cereal grains and seeds tend to maintain stable Mg levels even under varying Mg availability [44].

Plants regulate magnesium (Mg) distribution to support vital functions, transporting it through roots, vacuoles, and xylem. Mg is absorbed via specialized transporters and stored in vacuoles (20–120 mM concentrations), with its mobility facilitated by complexation with organic anions. Mg moves through apoplastic and symplastic pathways in roots, depending on transpiration and plant needs, before reaching shoots for photosynthesis and other processes. Most Mg remains in the symplastic pathway in shoots, accumulating in leaf cells. Research continues to explore Mg transport mechanisms to enhance plant health and crop yields [45,46,47].

The study showed variations in phosphorus (P) concentration among plant parts. Stems had the highest total P concentration (5.06 µg/mL), while roots contained the highest labile P (312.2 µg/mL), followed by leaves (98.44 µg/mL) and seeds (91.51 µg/mL). In stems, most P was stored in a complex form (1.96 µg/g), differing from other tissues. Phosphorus is challenging for plants to absorb due to its poor solubility and reliance on diffusion, a slow process supplying over 90% of a plant’s P needs. Soil factors, including moisture, temperature, and clay mineral types, significantly affect P mobility, with increased moisture enhancing diffusion and colder temperatures slowing the process. Soil composition also impacts how tightly P binds to particles, influencing its availability.

Plants mainly absorb P in two small forms: HPO_4_^2−^ (hydrogen phosphate) and H_2_PO^4−^ (dihydrogen phosphate), allowing them to pass through cell membranes. Larger phosphorus molecules cannot directly enter the cell membrane but can enter the space outside the cell membrane (apoplast), where they might transform to become more soluble. Once phosphorus is in a soluble form, it can be absorbed by plant cells. Some phosphorus is used by root cells directly, while most are transported to the inner part of the root (stele) through a specialized process called symplastic transport.

Calcium (Ca) concentrations varied among plant parts, with stems showing the highest levels (514 to 5305 µg/g) and seeds the lowest. Ca distribution in plants is tightly regulated through the xylem, which transports water and nutrients. This transport depends on the plant’s capacity and transpiration rate, both influenced by soil calcium availability and water loss through leaves. Ca moves in the xylem as free Ca^2+^ ions or complexes with organic acids like malate and citrate, with labile results indicating that most Ca is transferred as free ions. Ca can be transported within the xylem in two forms: as free Ca^2+^ ions or as complexes with organic acids like malate and citrate. The labile results indicate that calcium ions are transferred as free ions [48,49].

Potassium K is an essential nutrient that is absorbed by plants in more significant amounts than any other nutrient except N [50]. K is different from nitrogen in incorporation into the structures of organic compounds; instead, potassium remains in ionic form (K+) in solution in the cell and acts as an activator of many cellular enzymes [6]. Therefore, it has many functions in plant nutrition and growth that influence crop yield and quality. *B. elegans* has the second highest total concentration and labile of K in roots 915.0, 1329, leaves 784.0, 1355.2, seeds 639.0, 1032, and stems 285.75, 765.3 µg/g, respectively. K and Mg are critical in photosynthesis, carbohydrate partitioning, and protein synthesis.

### 3.4. Micronutrient Elements

The microelements analysis was done on four tissue plants obtained from *B. elegans*; the results of the analysis are presented in Table 5 and Figure 3.

Copper accumulation was predominantly observed in roots, with lower levels detected in leaves and no significant presence in stems or seeds. Roots had the highest total Cu concentration (8.46 µg/mL), while seeds exhibited the highest labile Cu concentration (49.58 µg/mL). No complex Cu fractions were detected in seeds or stems. Stems showed a total Cu concentration (4.77 µg/mL) similar to that of leaves but had a significantly higher labile Cu concentration (182.1 µg/mL) and no detectable complex Cu fraction. These findings suggest that Cu distribution within the plant is tissue-specific, with seeds potentially storing Cu, leaves containing a mix of labile and complex Cu, and stems primarily accumulating labile Cu.

Copper (Cu) distribution in plants shows most Cu concentrated in the rhizodermis and root cap, with lower levels in the inner cortex and stele, consistent with Cu’s affinity for cell walls via polygalacturonic acid-binding. In roots and leaves, Cu predominantly exists as sulfur-coordinated Cu (I) species linked to glutathione/cysteine-rich proteins. Isotopic variations indicate reductive uptake and redox cycling during translocation. Cu uptake is facilitated by transporters like the COPT and CTR families, while chaperone proteins (e.g., HMA and CCS families) direct Cu to specific cellular compartments [19,51,52].

In the soil, copper (Cu) exists in various forms with an average concentration ranging from 6 to 80 mg kg^−1^. Maintaining copper homeostasis in plants requires complex regulatory processes, with multiple transporters playing key roles. Among these, the copper transporter protein (COPT) family has been identified in plants based on sequence similarities with the Ctr protein or through functional complementation in yeast. Members of the COPT family are characterized by three predicted transmembrane segments, with most featuring an N-terminal region rich in methionine and histidine residues, indicative of potential metal-binding domains. Copper ions are uniquely capable of binding small molecules like O_2_ as ligands, making them essential cofactors for various oxidases, including mitochondrial cytochrome oxidase. These transporters often operate within a network, forming complexes to enhance their functionality. Their activity is influenced by factors such as copper availability and the presence of other metals, enabling plants to regulate copper levels effectively and prevent either deficiency or toxicity [53,54].

Iron (Fe) is essential for chlorophyll synthesis and photosynthesis, with younger leaves requiring more Fe than older ones. Fe distribution in plants varies significantly across tissues. Seeds had the lowest labile Fe concentration (0.81 µg/mL) but most Fe in a complex form (39.295 µg/g). Roots had the highest total Fe concentration (153.3 µg/mL) but a low labile fraction (6.55 µg/mL), while leaves showed intermediate total Fe levels (126.85 µg/mL), entirely in complex form. This distribution suggests seeds store complex Fe, while roots and leaves primarily accumulate Fe in complex forms.

Nitrogen form also influences Fe accumulation. Ammonium (NH_4_^+^) increases medium acidity, enhancing Fe solubility and promoting higher Fe concentrations in young leaves, as seen in petunia plants with greener foliage. Conversely, nitrate (NO_3_^−^) supply causes more Fe accumulation in roots and iron deficiency (chlorosis) in young leaves, as observed in corn plants. These patterns highlight the interplay between nitrogen forms and Fe distribution in plants.

Manganese is a vital micronutrient in plant health, playing a crucial role in light-induced water oxidation during photosynthesis and ATP synthesis. The data reveals manganese (Mn) concentration variations across different plant parts. Seeds contained the lowest labile Mn concentration (1.94 µg/mL), with most Mn existing in a complex form (13.23 µg/mL). Leaves exhibited a higher total Mn concentration (36.715 µg/g) than seeds, with a significant portion (25.28 µg/mL) found in the labile fraction. Stems also showed a moderate total Mn concentration (20.165 µg/mL) with a nearly equal distribution between labile (10.04 µg/mL) and complex Mn (10.125 µg/mL). Interestingly, roots had the highest total Mn concentration (9.435 µg/mL) but a lower labile Mn fraction (3.14 µg/mL) than leaves. Plants primarily absorb Mn^2+^ cations directly from soil solutions, which are transported to aerial plant parts in this form. Mn is further transported within plant tissues through xylem complexed with specific proteins or non-protein amino acids. The efficiency of Mn absorption depends on its transfer across the soil–root interface and the total amount of bioavailable Mn in the soil.

The interaction between manganese (Mn) and iron (Fe) in plant absorption varies across species. Fe reduces Mn toxicity by limiting its excessive uptake and accumulation rather than by increasing Fe^2+^ absorption or its corrective action within the plant. Research indicates that high Mn concentrations in barley seeds can promote plant growth and boost grain yield in Mn-deficient soils.

This study highlights Nickel (Ni) distribution patterns in plants, with seeds showing the highest labile Ni concentration (3.23 µg/mL) but the lowest total concentration (2.5 µg/mL). Leaves contain only complex Ni (1.885 µg/g), while stems have higher labile Ni (7.52 µg/mL) and total Ni (2.755 µg/mL). Roots show total Ni concentration (2.115 µg/mL), all in complex form. Ni uptake occurs primarily through roots, influenced by soil properties and the rhizosphere. Competing cations (e.g., Fe^2+^, Cu^2+^) can hinder Ni absorption. Foliar Ni concentrations in non-contaminated soil typically range from 0.05 to 10 µg/mL dry weight.

The study highlights variations in Zinc (Zn) distribution across plant tissues. Seeds have the highest labile Zn content, while leaves show a higher total Zn concentration but less labile Zn. Stems have the lowest total Zn content but a notable labile Zn fraction, with no complex forms detected. Roots have the second-highest total Zn content, entirely in the labile form. These patterns suggest seeds use labile Zn for germination, leaves balance Zn forms for functions and stems’ high labile Zn levels merit further investigation.

Zn is primarily absorbed as Zn^2+^ ions through roots, with some grasses also utilizing Zn–phytosiderophore (Zn–PS) pathways. Once absorbed, Zn is transported to shoot tissues via symplast, apoplast, and phloem pathways. Healthy leaves typically contain 15–100 mg Zn per kg dry weight, while Zn-deficient plants exhibit much lower levels. Zn tolerance does not directly correlate with low leaf Zn levels, as Zn distribution within leaves depends on the plant’s overall Zn status. Understanding these processes is essential for maintaining optimal Zn levels without surpassing safe thresholds.

### 3.5. Beneficial Elements

Table 6 and Figure 4 illustrate the findings on the total content, labile fractions, and complex fractions of beneficial elements.

Selenium (Se) is an essential micronutrient absorbed by plants primarily as inorganic selenates, which are converted into organic forms like selenocysteine (SeCys) and selenomethionine (SeMet). Its availability varies with soil pH, existing as selenate in alkaline soils and selenite in acidic soils, each differing in mobility and plant absorption.

The study shows that seeds have the highest total Se concentration (6.93 µg/mL), mostly in complex form. Leaves and stems have lower Se concentrations (2.505 µg/mL and 2.495 µg/mL, respectively), with Se in leaves entirely labile and stems evenly split between labile and complex forms. Roots contain the lowest total Se concentration (6.265 µg/mL) but a higher labile fraction. The mobility of Se is confirmed by the presence of selenite and selenomethionine in leaves, suggesting stems act primarily as a conduit between roots and leaves [55,56]

Sodium (Na) distribution varies among plant parts. Roots exhibited the highest labile Na concentration (584 µg/mL), and seeds had 97.37 µg/mL, both with no complex Na fraction identified, indicating a readily available pool of Na in these tissues. Leaves showed a very low total Na concentration (1.17 µg/mL), all in the labile form. Stems had a higher total Na concentration among above-ground parts (48.64 µg/mL), with all Na in the labile fraction as well.

While both sodium and potassium (K) are abundant alkali metals and monovalent cations, they serve different roles in plants. Potassium is an essential nutrient crucial for charge balance, enzyme activation, and osmotic regulation due to its larger ionic diameter and smaller hydration shell, which make it more suitable for these functions [57]. Sodium can be beneficial or toxic depending on the plant species and environmental conditions. In saline environments, plants must balance Na^+^ and K^+^ uptake to prevent Na toxicity and ensure sufficient K^+^ levels. High K^+^ concentrations can suppress Na^+^ uptake in some halophytes [46]. Understanding the interplay between Na and K is vital for maintaining plant health, especially in high-salinity areas [57,58].

The search results indicate that Cobalt is absorbed from the soil by plant roots and then translocated and distributed to other plant parts like stems and leaves. Cobalt concentrations can vary significantly across different plant species, ranging from 0.28 to 0.62 µg/mL. No complex Co fraction was detected in leaves, stems, and roots. In contrast, Co fraction was detected as complex in seed concentration 0.11 µg/mL.

Cobalt is an essential component of cobalamin, which is needed for the activities of several enzymes and coenzymes and is responsible for forming leghemoglobin, which is involved in nitrogen fixation in nodules of leguminous plants. Cobalt has both beneficial and harmful effects on plants. A relatively lower concentration of cobalt helped in better modulation and, consequently, better growth and yield. However, a higher level of cobalt reduced the bacterial population in the rhizosphere, and as a result, nodulation was hampered, leading to lower crop growth and yield [59].

### 3.6. Principal Component Analysis (PCA)

The PCA biplot provides a detailed and comprehensive visualization of the relationships between the concentrations of various elements (active variables) and different plant tissues (active observations). The elements analyzed include Pb, Co, Zn, Cd, Mn, Fe, Al, Mg, Ca, Na, P, Ni, Cr, K, Cu, As, and Se. These elements are represented as red vectors, with their direction and length indicating their contributions and influence on the two principal components, F1 (49.23%) and F2 (38.58%), which together explain 87.81% of the total variance in the dataset. The plant tissues—leaf, stem, root, and seed—are distributed across the plot based on their elemental composition.

Chemometric analysis was used in this study to analyze the eighteen elements in different tissues of **B. elegans** by reducing the dimensionality of the data and summarizing the relationships in graphs. This method provided an evaluation of the main components affecting the samples. As illustrated in Figure 5, leaves were closely associated with elements such as Zn, Cd, and Mn, reflecting higher concentrations of these elements in the leaf tissue. In contrast, stems aligned with elements like Ca, Na, and P and were positioned in the negative F1 and F2 quadrants. The root and seed were clustered together near elements such as Cu, As, and Se, indicating similarities in their elemental profiles, particularly for potentially toxic elements.

The scatter of data points in the PCA biplot indicates variation in the mineral element concentrations among leaf and stem samples. This variability may be attributable to biological factors, such as genetics, or environmental factors, such as soil composition. Roots and seeds, on the other hand, exhibit greater similarity in their mineral compositions compared to leaves and stems, likely due to their shared function as storage organs. Roots serve as reservoirs for essential nutrients and water, while seeds provide nourishment for the developing embryo [60]. This common role in resource storage likely explains their similar mineral profiles, contrasting with the distinct elemental distributions observed in leaves and stems. This analysis effectively highlights the nutrient allocation and potential toxicity levels within the plant, providing valuable insights into its elemental composition.

## 4. Conclusions and Future Perspectives

The study identified distinct distribution patterns of macronutrients, micronutrients, beneficial elements, and toxic metals in various parts of *B. elegans* (seeds, leaves, stems, and roots). The seeds contained the highest magnesium, phosphorus, selenium, and labile zinc levels. The leaves showed high calcium, manganese, iron, and total zinc concentrations. The roots accumulated significant amounts of potassium, copper, and arsenic, while the stems had elevated sodium and labile nickel levels. Toxic elements like cadmium and lead were primarily found in the leaves, whereas arsenic was concentrated in the roots. This compartmentalization underscores the plant’s capacity to regulate and distribute elements according to each tissue’s specific roles and requirements.

Future research on *Boerhavia elegans* should focus on several key areas to enhance understanding of its physiology and potential applications. First, investigating the molecular mechanisms behind the observed element distribution patterns in *B. elegans* is essential. Additionally, exploring the plant’s potential for phytoremediation, particularly in the context of arsenic-contaminated soils, could reveal new environmental applications. Assessing the nutritional value and safety of *B. elegans* seeds for human consumption is another important area, given their rich concentration of essential nutrients. Moreover, studies should examine the plant’s adaptability to diverse environmental conditions and its potential for cultivation in marginal lands. Investigating the effects of different cultivation practices on the elemental composition of *B. elegans* tissues is also crucial. Finally, comparing the elemental distribution patterns of *B. elegans* with those of other species in the Brassica genus may uncover unique traits or shared characteristics. These future research directions will provide valuable insights into *B. elegans* and its potential role in agriculture, environmental management, and addressing global food security challenges.

## Figures and Tables

**Figure 1 toxics-13-00014-f001:**
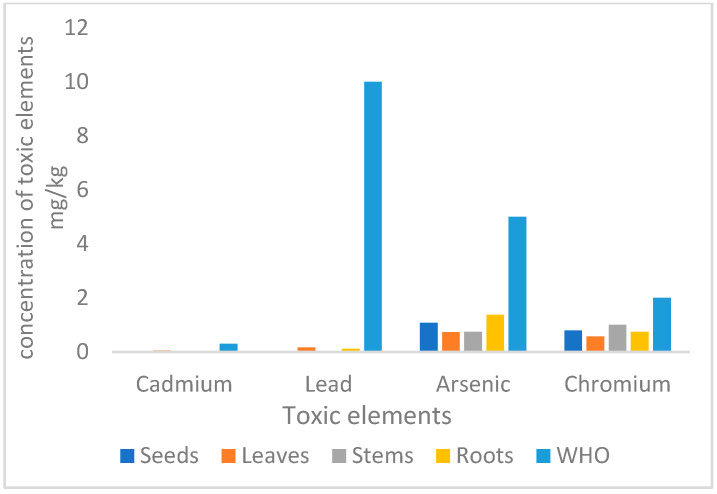
The concentration of cadmium, lead, and arsenic in *B. elegans* mg/kg.

**Figure 2 toxics-13-00014-f002:**
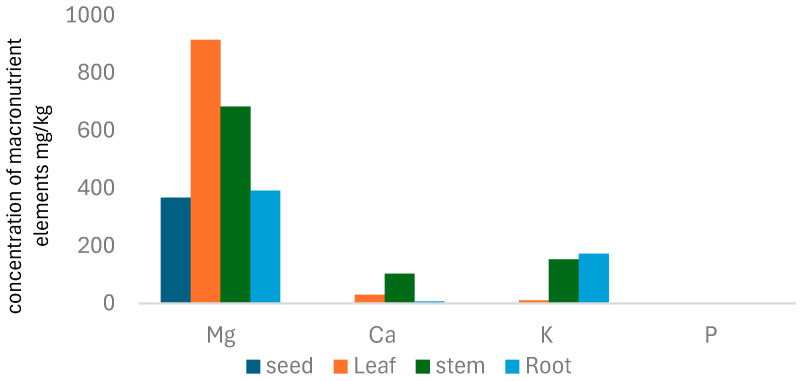
The concentration of macronutrient elements mg/kg in *B. elegans*.

**Figure 3 toxics-13-00014-f003:**
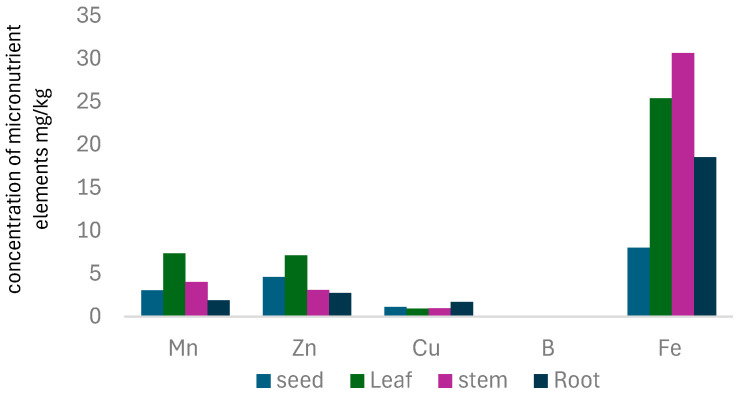
The concentration of micronutrient elements mg/kg in *B. elegans*.

**Figure 4 toxics-13-00014-f004:**
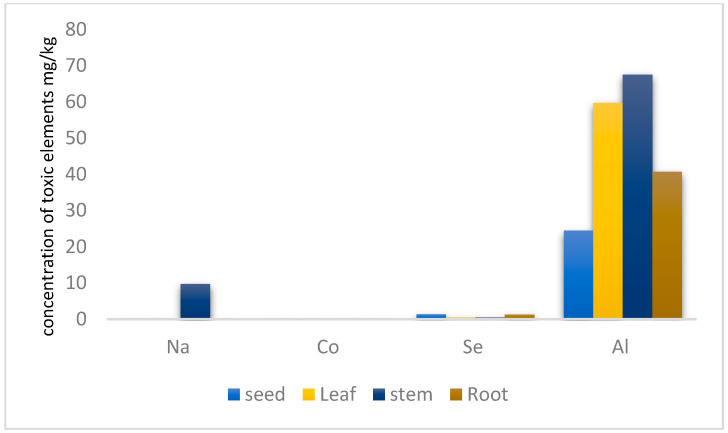
The concentration of the beneficial element mg/kg in *B. elegans*.

**Figure 5 toxics-13-00014-f005:**
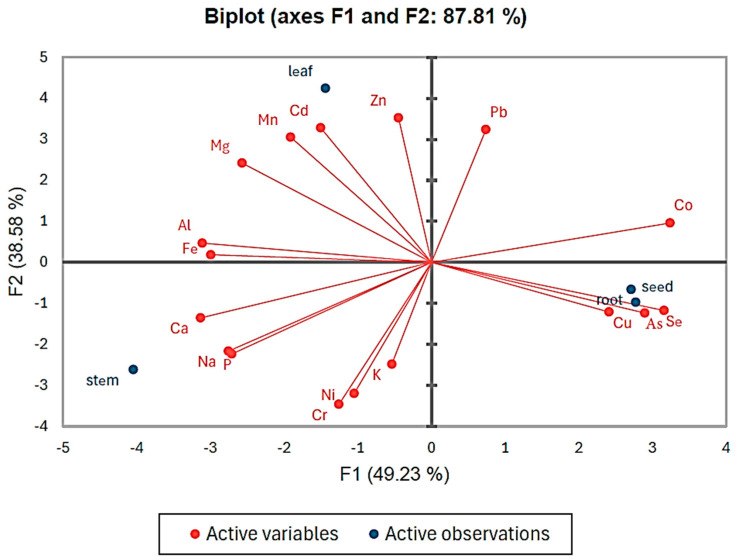
Biplot of principal component analysis.

**Table 1 toxics-13-00014-t001:** Mean concentration mg/L of trace metals in the tissue of *B. elegans* using ICP-OES.

Tissue	Seeds		Leaves		Stems		Roots	
Elements	HNO_3_	H_2_O	HNO_3_	H_2_O	HNO_3_	H_2_O	HNO_3_	H_2_O
Cd	0.009 ± 0.0003	0 ± 0.0012	0.054 ± 0.0019	0 ± 0.0007	0.021 ± 0.0049	0 ± 0.0014	0.018 ± 0.0002	0.061 ± 0.0025
Co	0.124 ± 0.0009	0.051 ± 0.0011	0.097 ± 0.0004	0.08 ± 0.0001	0.056 ± 0.0009	0.061 ± 0.0009	0.118 ± 0.0004	0.059 ± 0.0008
Cr	0.793 ± 0.0025	0 ± 0.0003	0.565 ± 0.003	0 ± 0.0006	1.004 ± 0.001	0 ± 0.0001	0.738 ± 0.0014	0 ± 0.0003
Cu	1.13 ± 0.0052	4.958 ± 0.0354	0.91 ± 0.0066	0.207 ± 0.0019	0.954 ± 0.0189	18.21 ± 0.053	1.692 ± 0.0126	0.292 ± 0.0008
Fe	8.021 ± 0.0653	0.081 ± 0.0015	25.37 ± 0.15	0 ± 0.0004	30.66 ± 0.083	0.655 ± 0.0045	18.54 ± 0.13	0 ± 0.0004
Mg	367.8 ± 0.75	14.5 ± 0.121	914.7 ± 3.89	259.9 ± 1.32	683.4 ± 2.85	102 ± 0.33	392.2 ± 2.91	59.84 ± 0.351
Mn	3.034 ± 0.0091	0.194 ± 0.0003	7.343 ± 0.0314	1.264 ± 0.0062	4.033 ± 0.0449	1.004 ± 0.0141	1.887 ± 0.0119	0.157 ± 0.0007
Ni	0.5 ± 0.0019	0.323 ± 0.002	0.377 ± 0.0026	0 ± 0.0012	0.551 ± 0.0023	0.752 ± 0.0037	0.423 ± 0.0027	0 ± 0.0007
Pb	0.033 ± 0.0061	0.282 ± 0.0009	0.165 ± 0.0053	0 ± 0.0078	0 ± 0.0162	1.333 ± 0.0061	0.113 ± 0.0078	0.04 ± 0.0048
Al	24.49 ± 2.154	0 ± 0.0018	59.71 ± 1.255	0 ± 0.002	67.53 ± 1.062	0 ± 0.0059	40.67 ± 0.426	0 ± 0.0079
As	1.068 ± 0.1653	0.109 ± 0.0264	0.728 ± 0.0407	0.177 ± 0.0252	0.742 ± 0.0164	0.099 ± 0.0138	1.374 ± 0.0185	0.221 ± 0.0306
Se	1.386 ± 0.1493	0.126 ± 0.0134	0.501 ± 0.0063	0.147 ± 0.0149	0.499 ± 0.0072	0.119 ± 0.0128	1.253 ± 0.0375	0.258 ± 0.0191
Zn	4.619 ± 0.1817	34.52 ± 0.194	7.117 ± 0.0995	1.279 ± 0.0059	3.108 ± 0.0453	64.03 ± 0.37	2.754 ± 0.0035	0.643 ± 0.0039
Na	0 ± 0.0128	9.737 ± 0.2996	0.234 ± 0.1354	30.3 ± 5.058	9.728 ± 1.3931	50.68 ± 1.328	0 ± 0.0385	29.2 ± 0.61
K	2.649 ± 0.0068	103.2 ± 0.37	9.937 ± 0.6758	67.76 ± 0.064	152.9 ± 6.95	76.53 ± 3.66	173 ± 5.24	66.46 ± 0.357
P	0 ± 0.0031	9.151 ± 0.2234	0 ± 0.026	4.922 ± 0.3786	1.012 ± 0.8824	0.31 ± 0.0252	0 ± 0.0032	15.61 ± 0.279
B	0 ± 0.0012	0.714 ± 156	0 ± 0.0012	2.669 ± 2051	0 ± 0.0041	0.173 ± 0.0184	0 ± 0.0003	0.244 ± 0.004
Ca	1.061 ± 0.0501	20.99 ± 0.149	30.17 ± 0.0501	255.4 ± 3.61	102.8 ± 7.08	8.533 ± 0.0554	6.939 ± 0.8491	1.38 ± 0.0587

Mean ± Standard Deviation.

**Table 2 toxics-13-00014-t002:** Variance of element concentrations across plant tissues.

Groups	Count	Sum	Average	Variance
seed	18	2083.585	115.7547	185,790.4
leaves	18	5289.89	293.8828	1,146,789
stems	18	5294.99	294.1661	653,386
roots	18	3208.595	178.2553	239,611.8

**Table 3 toxics-13-00014-t003:** Total concentration µg/mL, labile µg/mL, and complex fraction µg/mL in *B. elegans* using wet and labile digestion toxic element.

Tissue	Seed			Leaves			Stem			Root		
Elements	Total Concentration	Labile	Complex Fraction	Total Concentration	Labile	Complex Fraction	Total Concentration	Labile	Complex Fraction	Total Concentration	Labile	Complex Fraction
cd	0.045	0	ND	0.27	0	ND	0.105	0	ND	0.09	1.22	ND
pb	0.165	2.82	ND	0.825	0	ND	0	13.33	ND	0.565	0.8	ND
As	5.34	1.09	4.25	3.64	3.54	0.1	3.71	0.99	2.72	6.87	4.42	2.45
Cr	3.965	0	ND	2.825	ND	ND	5.02	ND	6.01	3.69	0	ND

ND = not detect.

**Table 4 toxics-13-00014-t004:** Total concentration µg/mL, labile µg/mL, and complex fraction µg/mL in *B. elegans* using wet and labile digestion of macronutrient element.

Tissue	Seed			Leaves			Stem			Root		
Elements	Total Concentration	Labile	Complex Fraction	Total Concentration	Labile	Complex Fraction	Total Concentration	Labile	Complex Fraction	Total Concentration	Labile	Complex Fraction
Mg	1839	145	1694	4573.5	5198	ND	3417	1020	2397	1961	1196.8	764.2
Ca	5.305	209.9	ND	150.85	5108	ND	514	85.33	428.67	34.695	27.6	7.095
K	13.245	1032	ND	49.685	1355.2	ND	764.5	765.3	ND	865	1329.2	ND
P	0	91.51	ND	0	98.44	ND	5.06	3.1	1.96	0	312.2	ND

ND = not detect.

**Table 5 toxics-13-00014-t005:** Total concentration µg/mL, labile µg/mL, and complex fraction µg/mL in *B. elegans* using wet and labile digestion of micronutrient elements.

Tissue	Seed			Leaves			Stem			Root		
Elements	Total Concentration	Labile	Complex Fraction	Total Concentration	Labile	Complex Fraction	Total Concentration	Labile	Complex Fraction	Total Concentration	Labile	Complex Fraction
Cu	5.65	49.58	ND	4.55	4.14	0.41	4.77	182.1	ND	8.46	5.84	2.62
Fe	40.105	0.81	39.295	126.85	0	126.85	153.3	6.55	146.75	92.7	0	92.7
Mn	15.17	1.94	13.23	36.715	25.28	11.435	20.165	10.04	10.125	9.435	3.14	6.295
Zn	23.095	345.2	ND	35.585	25.58	10.005	15.54	640.3	ND	13.77	12.86	0.91
Ni	2.5	3.23	ND	1.885	0	1.885	2.755	7.52	ND	2.115	0	2.115
B	0	7.14	ND	0	53.38	ND	0	1.73	ND	0	4.88	ND

ND = not detect.

**Table 6 toxics-13-00014-t006:** Total concentration µg/mL, labile µg/mL, and complex fraction µg/mL in *B. elegans* using wet and labile digestion of beneficial elements.

Tissue	Seed			Leaves			Stem			Root		
Elements	Total Concentration	Labile	Complex Fraction	Total Concentration	Labile	Complex Fraction	Total Concentration	Labile	Complex Fraction	Total Concentration	Labile	Complex Fraction
Na	0	97.37	ND	1.17	606	ND	48.64	506.8	ND	0	584	ND
Co	0.62	0.51	0.11	0.485	1.6	ND	0.28	0.61	ND	0.59	1.18	ND
Se	6.93	1.26	5.67	2.505	2.94	ND	2.495	1.19	1.305	6.265	5.16	1.105
Al	122.45	0	122.45	298.55	0	298.55	337.65	0	337.65	203.35	0	203.35
Na	0	97.37	ND	1.17	606	ND	48.64	506.8	ND	0	584	ND

ND = Not detected.

## Data Availability

The data presented in this study are available on request from the corresponding author.

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
