# Peer review of "Trace Element Speciation and Nutrient Distribution in Boerhavia elegans: Evaluation and Toxic Metal Concentration Across Plant Tissues"

_toxics, 2024, doi:10.3390/toxics13010014_

Round 1
Reviewer 1 Report
Comments and Suggestions for Authors
Toxics3342007
Title: Trace Element Speciation and Nutrient Distribution in Boerhavia elegans: Evaluation and Toxic Metal Concentration Across plant Tissues
Authors: Tahreer M. et al.
General Comments
The paper examines nutrient element concentrations in tissue from a medicinal plant, Boerhavia elegans. The idea is okay. However, the paper is much longer than need be, as it contains general text that applies to any plant. Either the text should be specific to Boerhavia elegans or be a review of nutrients in plant tissue.
For example, if the focus is on Boerhavia elegans, you need to compare concentrations of the elements to other plants. Does Boerhavia elegans have especially large or small concentrations relative to other plants species? This interpretation of the findings is missing.
Although the focus is supposed to be on toxic metals, this starts on page 14. Why is it relegated to the end of the paper?
Overall, the text could be reduced by more than 50%, and the paper would still be readable.
Another substantial comment is that the statistical analyses are lacking. The tables and figures with data need to show the differences that are statistically significant.
Also, you need to be careful with wording. The values are concentrations not content. Thus, statements like ‘seeds store X nutrient’ is not necessarily correct. They might have a greater concentration, but the ‘storage’ depends on mass of the tissue. Seed mass is trivial compared to the mass of leaves, shoots, and roots.
Indeed, change ‘content’ to ‘concentration’ throughout the manuscript.
Specific Comments
1) Thank you for defining ‘heavy metals’ on line 64, rather than assume all trace metals are ‘heavy.’
2) The Methods, especially sampling need more detail. Tissue nutrient concentration often vary with plant age. How old were the plants when collected?
3) The sample sizes are not clear to me.
4) The Statistical Method section needs some work. There is no need to mention that you used excel for summary statistics. Rather describe how you determined whether differences were statistically significant or not. Did you use Analysis of Variance? Describe the model used to determine differences.
5) The Results and Discussion section could use some reorganization. For example, there is no need to start with general information that applies to all plant studies. In other words, delete lines 185 to 207. Start with the specific findings in your study. In other words, describe the statistically significant findings in Table 1.
6) You need to describe the statistically significant differences in Tables 2, 3, and 4. Simply presenting the findings, on line 233, is too terse.
7) The text on page 10 and 13 is interesting but most of it seems to be directly taken from a plant physiology textbook. Sorry to be harsh, but none of this is new information.
8) The heart of the paper starts on page 15 with potentially toxic elements. Are the concentrations in Boerhavia elegans near toxic levels?
9) Sorry to be harsh, but Figure 5 is the most interesting part of the paper.
10) Some of the references are incomplete, cf., number 12.
Technical Comments
1) Line 122: ‘specimens’ is not clear. Do you mean the four tissue types? Or four replicates of each tissue type?
2) Line 125: provide more detail on sampling roots, stems, and leaves. Were these collected from field plots?
3) Line 144 – 148: if the wavelengths are different from standard then there is no need to include, unless you used unique ones for detection.
4) Line 182: also describe the variables included in the PCA analysis. What was the question you were trying to answer.
5) Table 1: what are the concentration units?
6) Lines 210 to 225: too general and could be deleted.
7) Lines 226 to 233: reduce to one sentence.
8) Figures 1, 2, 3, and 4: could be better presented. The x-axis categories should be nutrient, not tissue type. The important comparison is how concentrations differed among tissue types for a chemical element, not differences between chemical elements for a tissue type.
9) Table 2, Figure 1: do these show the same data? It is difficult for me to assess.
10) Table 2: consider ‘not extractable’ rather than ‘complex fraction.’ You do not know where it is, but that it is not water extractable.
11) Line 258: I doubt that plants can ‘meticulously regulate’ anything. They have transports that might be specific for chemical elements, but regulation is a complex process, not meticulous. Start by reading, Chen, Z.C. and Ma, J.F., 2013. Magnesium transporters and their role in Al tolerance in plants. Plant and Soil, 368, pp.51-56.
12) Line 284: delete ‘microscopic.’
13) Table 3, Figure 2: do these show the same data? It is difficult for me to assess.
14) Line 298: ‘calcium’ or ‘Ca’ but not both.
15) Line 336: sorry but you lost. How is this evident in Table 1?
16) Line 339: again, you lost me. I am not aware that ICP-OES can identify organic compounds and binding.
17) Line 340: is this in your methods?
18) Line 344 & 345: describe the COPT/CTR family of proteins for naïve readers, cf., Yuan, M., Li, X., Xiao, J. and Wang, S., 2011. Molecular and functional analyses of COPT/Ctr-type copper transporter-like gene family in rice. BMC plant biology, 11, pp.1-12.
19) Line 348: your findings or literature values?
Author Response
Reviewer 1
General Comments
The paper examines nutrient element concentrations in tissue from a medicinal plant, Boerhavia elegans. The idea is okay. However, the paper is much longer than needed, as it contains general text that applies to any plant. Either the text should be specific to Boerhavia elegans or be a review of nutrients in plant tissue. For example, if the focus is on Boerhavia elegans, you need to compare concentrations of the elements to other plants. Does Boerhavia elegans have especially large or small concentrations relative to other plants species? This interpretation of the findings is missing.
I understand your feedback and will summarize the paper by 50%, ensuring it is specific to the plant Boerhavia elegans while removing the excessive general text.
Although the focus is supposed to be on toxic metals, this starts on page 14. Why is it relegated to the end of the paper?
I respect your feedback and will rearrange the content to make sure toxic metals are addressed earlier in the paper.
Overall, the text could be reduced by more than 50%, and the paper would still be readable.
I respect your feedback, and I will work on reducing the paper's length by more than 50% while ensuring it remains clear and readable.
Another substantial comment is that the statistical analyses are lacking. The tables and figures with data need to show the differences that are statistically significant.
Also, you need to be careful with wording. The values are concentrations not content. Thus, statements like ‘seeds store X nutrient’ is not necessarily correct. They might have a greater concentration, but the ‘storage’ depends on mass of the tissue. Seed mass is trivial compared to the mass of leaves, shoots, and roots.
Indeed, change ‘content’ to ‘concentration’ throughout the manuscript.
I will make sure to replace all instances of "content" with "concentration" throughout the manuscript.
Specific Comments
1) Thank you for defining ‘heavy metals’ on line 64, rather than assume all trace metals are ‘heavy.’
Thank you, I truly appreciate your acknowledgment!
2) The methods, especially sampling need more detail. Tissue nutrient concentration often vary with plant age. How old were the plants when collected?
3) The sample sizes are not clear to me.
Thank you for your feedback. I will revise the methods section to provide more clarity and include detailed information about the sampling process. Since this plant can be either annual or perennial depending on environmental conditions, determining its exact age is challenging due to its growth pattern.
4) The Statistical Method section needs some work. There is no need to mention that you used Excel for summary statistics. Rather describe how you determined whether differences were statistically significant or not. Did you use Analysis of Variance? Describe the model used to determine differences.
5) The Results and Discussion section could use some reorganization. For example, there is no need to start with general information that applies to all plant studies. In other words, delete lines 185 to 207. Start with the specific findings in your study. In other words, describe the statistically significant findings in Table 1.
I respect your feedback and have deleted lines 185 to 207 in the text. I will reorganize the Results and Discussion section to start with the specific, statistically significant findings from Table 1.
6) You need to describe the statistically significant differences in Tables 2, 3, and 4. Simply presenting the findings, on line 233, is too terse.
Thank you for your feedback. I will delete the mentioned line and provide a more detailed description of the statistically significant differences in Tables 2, 3, and 4 within the text to ensure clarity and completeness.
7) The text on pages 10 and 13 is interesting but most of it seems to be directly taken from a plant physiology textbook. Sorry to be harsh, but none of this is new information.
I understand your feedback, and I appreciate your honesty. While the text may seem like standard information, it serves to explain and discuss the mechanisms of element transfer, which are relevant to the context of the study. However, I will review and revise it to ensure it adds unique value and aligns with the focus of the paper.
8) The heart of the paper starts on page 15 with potentially toxic elements. Are the concentrations in Boerhavia elegans near toxic levels?
Thank you for your question. The concentrations in Boerhavia elegans are detailed in line 534, where the levels are discussed about their potential toxicity.
9) Sorry to be harsh, but Figure 5 is the most interesting part of the paper.
Thank you for your honest feedback. I appreciate your perspective, and I’m glad you found Figure 5 to be engaging. I’ll ensure it is given the emphasis it deserves in the paper.
10) Some of the references are incomplete, cf., number 12.
Thank you for pointing that out. I will review and complete the references, including number 12
Technical Comments
1) Line 122: ‘specimens’ is not clear. Do you mean the four tissue types? Or four replicates of each tissue type?
2) Line 125: provide more detail on sampling roots, stems, and leaves. Were these collected from field plots?
I will rewrite this section in lines 122 and 125, adding more details about the sampling process for roots, stems, and leaves, including whether they were collected from field plots.
3) Line 144 – 148: if the wavelengths are different from standard then there is no need to include unless you used unique ones for detection.
Thank you for your feedback. I have removed the wavelengths from lines 144–148
4) Line 182: also describe the variables included in the PCA analysis. What was the question you were trying to answer.
Thank you for your feedback. I will revise line 182 to include a clear description of the variables used in the PCA analysis and elaborate on the specific research question being addressed.
5) Table 1: what are the concentration units?
Thank you for pointing that out. I will add the concentration units to ensure clarity and consistency in the data presentation.
6) Lines 210 to 225: too general and could be deleted.
7) Lines 226 to 233: reduce to one sentence.
I will make the necessary revisions—deleting lines 210 to 225 and condensing lines 226 to 233 into a single sentence.
8) Figures 1, 2, 3, and 4: could be better presented. The x-axis categories should be nutrient, not tissue type. The important comparison is how concentrations differed among tissue types for a chemical element, not differences between chemical elements for a tissue type.
I will revise Figures 1, 2, 3, and 4 to better present the data by changing the x-axis categories to nutrients.
9) Table 2, Figure 1: do these show the same data? It is difficult for me to assess.
Thank you for your feedback. No, the data is not the same, but I will make changes to ensure it is presented more clearly.
10) Table 2: consider ‘not extractable’ rather than ‘complex fraction.’ You do not know where it is, but it is not water extractable.
Thank you for your feedback. The term "complex fraction" represents the non-free form of the element, which is certainly in a complexed state. Therefore, it was referred to as the "complex fraction," a term commonly used to describe non-free ions. However, I prefer the term "complex fraction," but I do not mind changing it if deemed necessary.
11) Line 258: I doubt that plants can ‘meticulously regulate’ anything. They have transports that might be specific for chemical elements, but regulation is a complex process, not meticulous. Start by reading, Chen, Z.C. and Ma, J.F., 2013. Magnesium transporters and their role in Al tolerance in plants. Plant and Soil, 368, pp.51-56.
Thank you for your feedback. I will revise and change the statement in line 258 to better reflect the complexity of plant regulation processes.
12) Line 284: delete ‘microscopic.’
Thank you for pointing that out. I will delete the word "microscopic" as suggested.
13) Table 3, Figure 2: do these show the same data? It is difficult for me to assess.
Thank you for your feedback. No, the data is not the same, but I will make changes to ensure it is presented more clearly.
14) Line 298: calcium’ or ‘Ca’ but not both.
Thank you for pointing that out. I will ensure consistency by using only one term throughout the text
15) Line 336: sorry but you lost. How is this evident in Table 1?
Thank you for your feedback. I will review Table 1 and ensure that the explanation in line 336 is clear and directly supported by the data presented.
16) Line 339: again, you lost me. I am not aware that ICP-OES can identify organic compounds and binding.
17) Line 340: is this in your methods?
Thank you for your feedback. It is not about the device's ability to detect, but rather that the distribution of copper in our study is similar to that observed in most plants. However, I will revise and rewrite the section to ensure it is clearer and more understandable.
18) Line 344 & 345: describe the COPT/CTR family of proteins for naïve readers, cf., Yuan, M., Li, X., Xiao, J. and Wang, S., 2011. Molecular and functional analyses of COPT/Ctr-type copper transporter-like gene family in rice. BMC plant biology, 11, pp.1-12.
19) Line 348: your findings or literature values?
Thank you for your feedback. I will add a brief description of the COPT/CTR family of proteins to lines 344 and 345, referencing the suggested study for clarity. Regarding line 348, I will clarify whether the reported values are based on my findings or derived from the literature.
Reviewer 2 Report
Comments and Suggestions for Authors
The abstract lacks a description of scientific issues. In addition, most of them are the results of the studies elaborated and lack clear conclusions.
The author needs to re-elaborate the sampling method and determine whether the samples collected in different regions are representative and comparable. Even within the same area, the sampling method needs to be described in detail.
Detection limit for heavy metals and relative error?
What is the basis of digestion using a solution of 65% HNO3 and 30% H2O2?
In the Results and Discussion section, the author describes several facts that already exist. These facts are clearly agreed upon by everyone, and such a large number of descriptions weaken the importance of this article. It is recommended to reorganize the Discussion section and focus on the theme of this article.
Author Response
The abstract lacks a description of scientific issues. In addition, most of them are the results of the studies elaborated and lack clear conclusions.
Thank you for your feedback. I will revise the abstract to include a clear description of the scientific issues and provide concise, well-defined conclusions alongside the study results.
The author needs to re-elaborate the sampling method and determine whether the samples collected in different regions are representative and comparable. Even within the same area, the sampling method needs to be described in detail.
Thank you for your feedback. I will rewrite and clarify the sampling method, ensuring it includes detailed descriptions and addresses whether the samples collected from different regions are representative and comparable, even within the same area.
Detection limit for heavy metals and relative error?
Thank you for your feedback. I will include an analysis of the variance in element concentrations across plant tissues to provide a more detailed understanding of the data, and I have added the standard deviation to Table 1
What is the basis of digestion using a solution of 65% HNO3 and 30% H2O2?
Thank you for your question. The percentages, 65% HNO3 and 30% H2O2, refer to the concentrations of the stock solutions used in the preparation of the digestion mixture. These percentages represent the purity of the nitric acid and hydrogen peroxide in their respective bottles, typically indicated by the manufacturer.
In the Results and Discussion section, the author describes several facts that already exist. These facts are clearly agreed upon by everyone, and such a large number of descriptions weaken the importance of this article. It is recommended to reorganize the Discussion section and focus on the theme of this article.
Thank you for your feedback. I will revise the Results and Discussion section to reduce redundant descriptions of well-established facts.
Round 2
Reviewer 1 Report
Comments and Suggestions for Authors
I reviewed the original. Thank you for addressing my comments and concerns.
Reviewer 2 Report
Comments and Suggestions for Authors
The author has modified the comments, and I agree to accept it.